# Preventing iatrogenic HCV infection: A quantitative risk assessment based on observational data in an Egyptian hospital

**Paul Henriot** [1,2] *, **Wagida A. Anwar** [3], **Maha El Gaafary** [3], **Samia Abdo** [4], **Mona Rafik** [4†], **Wafaa M. Hussein** [3], **Dalia Sos** [3], **Isis Magdy** [3], **Kévin Jean** [1,2], **Laura Temime** [1,2]

**1** MESuRS Laboratory, Conservatoire national des arts et métiers, Paris, France, **2** PACRI Unit, Conservatoire national des arts et métiers, Institut Pasteur, Paris, France, **3** Department of Community, Environmental and Occupational Medicine, Faculty of Medicine, Ain Shams University, Cairo, Egypt, **4** Department of Clinical Pathology, Faculty of Medicine, Ain Shams University, Cairo, Egypt

† Deceased.
* paul.henriot@protonmail.com

**Data Availability Statement:** The data analysed in this study is available upon request only. Indeed, de-identified data cannot be publicly shared, as our

## Abstract

When compliance with infection control recommendations is non-optimal, hospitals may play an important role in hepatitis C (HCV) transmission. However, few studies have analyzed the nosocomial HCV acquisition risk based on detailed empirical data. Here, we used data from a prospective cohort study conducted on 500 patients in the Ain Shams hospital (Cairo, Egypt) in 2017 with the objective of identifying (i) high-risk patient profiles and (ii) transmission hotspots within the hospital. Data included information on patient HCV status upon admission, their trajectories between wards and the invasive procedures they underwent. We first performed a sequence analysis to identify different hospitalization profiles. Second, we estimated each patient's individual risk of HCV acquisition based on ward-specific prevalence and procedures undergone, and risk hotspots by computing ward-level risks. Then, using a beta regression model, we evaluated upon-admission factors linked to HCV acquisition risk and built a score estimating the risk of HCV infection during hospitalization based on these factors. Finally, we assessed and compared ward-focused and patient-focused HCV control strategies. The sequence analysis based on patient trajectories allowed us to identify four distinct patient trajectory profiles. The risk of HCV infection was greater in the internal medicine department, compared to the surgery department (0·188% [0·142%-0·235%] vs. 0·043%, CI 95%: [0·036%-0·050%]), with risk hotspots in the geriatric, tropical medicine and intensive-care wards. Upon-admission risk predictors included source of admission, age, reason for hospitalization, and medical history. Interventions focused on the most at-risk patients were most effective to reduce HCV infection risk. Our results might help reduce the risk of HCV acquisition during hospitalization in Egypt by targeting enhanced control measures to ward-level transmission hotspots and to at-risk patients identified upon admission.

study involves sensitive data on human participants, and could be indirectly identifying based on multiple patient characteristics. Individual data requests may be sent to the CorC (secr-CORC@pasteur.fr).

**Funding:** PH was funded by Agence Nationale de Recherches sur le Sida et les Hépatites Virales (ANRS), Grant Number 12320 B115. The funder had no role in study design, data collection and analysis, decision to publish, or preparation of the manuscript.

**Competing interests:** The authors have declared that no competing interests exist.

# Introduction

Hepatitis C virus (HCV) is a bloodborne pathogen usually transmitted during iatrogenic procedures or unsafe injections like drug use. Even though a direct-acting antiviral (DAA) treatment is available, an estimated 58 million people were still living with hepatitis C worldwide in 2019, with significant morbidity and mortality consequences, mostly due to liver cirrhosis and hepatocellular carcinoma [1]. While HCV prevalence varies widely between countries, Egypt has historically been the most affected country worldwide, with an HCV prevalence still over 10% in the adult population in the mid 2010's [2]. The scaling-up of DAA treatments has temporarily yielded the hope that a large test-and-treat strategy could be sufficient to eliminate the epidemics [2]. In this context, Egypt launched an ambitious national treatment programme in 2014, followed by an intensive screening and treatment programme in 2018, with the objective of eliminating HCV within the country by 2021. In 2018 and 2019, almost 50 million Egyptian residents were screened (80% of the adult population), resulting in an HCV seroprevalence reduced to 4·6% in the adult population [3, 4]. It is estimated that more than 90% of the HCV infected population were treated during this campaign, and the 2021 general prevalence was estimated at around 0.5%, thus achieving WHO elimination target. Despite these huge achievements, 20% of the adult population were not screened and treatment does not prevent for reinfections, which is especially problematic in specific population such as injectable drug users, whom could introduce HCV in hospitals. Hence, articulating primary HCV prevention, especially in healthcare settings [5], together with treatment and cure remains key to accelerate HCV elimination in the country.

While the implementation of infection control measures has substantially reduced the risk of nosocomial HCV transmission, hospitals may still play an important role in the epidemic dynamics of HCV, due to potential exposure to infected patients and contaminated equipment [6, 7]. This is particularly true in low-to-middle-income countries, such as Egypt [8, 9], However, HCV outbreaks in healthcare settings are seldom detected and investigated, so that the transmission routes often remain unknown.

A few studies quantified the HCV acquisition risk in hospitals [10–16]. However, these studies were mostly focused on the occupational risk to healthcare workers, most used data from the literature rather than actual observations, and none accounted for procedure-specific risk levels. In this context, the first objective of this work is to assess the risk of nosocomial HCV infection for the patients hospitalized in an Egyptian hospital. To that aim, we propose a probabilistic risk assessment framework informed by detailed empirical data recently collected in this hospital. Based on this assessment, the second objective is then to identify transmission hotspots as well as at-risk patient profiles to better manage the HCV risk within the hospital.

# Material and methods

## Ethical considerations

Ethical approval was obtained from the Institutional Review Board of the Faculty of Medicine of Ain Shams University and from the Sheffield University, School of Health and Related Research. **All methods were carried out in accordance with relevant guidelines and regulations and informed consent was obtained from all subjects and/or their legal guardian(s).**

## Data and setting

*D*ata was collected as part of a prospective cohort study (ANRS 12320 IMMHoTHep project, "Investigative Mathematical Modeling of Hospital Transmission of Hepatitis C") conducted over a 6-month period in 2017 [17]. This study focused on patients hospitalized in the internal

**Table 1. Odds-ratios (OR) of HCV infection associated with exposure to iatrogenic procedures, based on a previously published meta-analysis [18].** The 15 procedure types in the IMMHoTHep data (second column) are aggregated into 8 of the 10 procedure groups defined in the meta-analysis and sorted from higher to lower risk. No procedures from the remaining 2 groups defined in the meta-analysis (dental care and transplantation) were observed in the IMMHoTHep data.

| Procedure groups in meta-analysis | Procedures in data | OR [CI 95%] |
|---|---|---|
| *Wound care* | *Stitches, Wound dressing* | 2.83 [1.85–4.32] |
| *Blood transfusion* | *Blood transfusion* | 2.60 [2.09–3.22] |
| *IV—Catheter* | *Intravenous, Cardiac catheter* | 2.42 [1.68–3.51] |
| *Surgery* | *Surgery* | 2.30 [1.77–3.00] |
| *Other procedures* | *Other invasive procedures, drainage catheter* | 2.28 [1.43–3.64] |
| *Haemodialysis* | *Dialysis* | 2.02 [0.98–4.17] |
| *Injection* | *Injection, Blood glucose, Blood sample* | 1.67 [1.17–2.38] |
| *Endoscopy* | *Endoscopy, Endotracheal intubation, gastric lavage* | 1.48 [0.95–2.3] |

medicine (organized in 15 wards) and surgery (organized in 10 wards) departments of the Ain Shams University Hospital in Cairo, Egypt.

Five hundred hospitalized patients (aged more than 21) were included upon their admission to the hospital, either through the outpatient clinics or the emergency departments. Their demographic characteristics and medical history were collected through a structured questionnaire upon admission. Their HCV status upon admission was retrieved, and infections were confirmed by HCV-RNA detection. Patients' individual trajectories were then followed up over the course of their entire hospitalizations: this included information on their geographical movements between departments and wards within the hospital and the invasive procedures they underwent within these locations. Procedures performed were aggregated into 15 groups following expert opinion (Table 1). Further information on the study is available in Anwar et al., (2021) [17].

## Trajectory analysis

To identify typical hospitalization profiles, we performed a sequence analysis based on patient trajectories between seven locations (Surgery department, Internal medicine department, Emergency room, ICU, Endoscopy building, MRI building, Outpatients clinic) within the hospital. Sequence analysis is a non-parametric approach to investigate and cluster longitudinal life course data between individuals [18, 19]. Here, sequences were composed of five-minute-long events over the course of hospitalization, completed by the post-hospitalization status (i.e., deceased or discharged), so that all trajectories had the same length as the longest one. Therefore, each sequence was composed of at least two out of nine states, describing location within the hospital (seven states) and post-hospitalization status (two states).

To compute differences between sequences, substitution and indel costs were calculated based on the observed transition rates between the states previously defined. We used the optimal matching (OM) method to compute the distance matrix between individual sequences [20]. Then, we compared partitions built with the Ward's minimum variance method [21] using the Point Biserial Correlation (PBC) [22] to find the optimal number of clusters.

All sequence analyses were performed using the R package TraMineR [23].

## Per-procedure risk estimation

We firstly estimated the risk of iatrogenic HCV infection following a procedure performed with contaminated equipment, for each of the 15 procedure types identified in the data. This

was based on a previous meta-analysis studying the association between HCV infection and ten groups of iatrogenic procedures [24]. The 15 procedure types in the data were aggregated to match the groups considered in this meta-analysis (Table 1). Odds-ratio (OR) distributions were considered log-normal with mean equal to the average ORs and standard deviation derived from the associated confidence intervals.

The risk of getting HCV-infected through injection by contaminated equipment was then used as a reference to determine the other procedure-specific risks. Ross et al. [11] estimated this risk at 2·20% (plausible interval, 1%-9·2%). Here, we translated this as a PERT-distributed [25] risk distribution, with a median of 2·20% and an analytically calculated mode of 1·23% (S1 Text):

$$Risk_{injection} \sim \mathrm{PERT}(1, 1 \cdot 23, 9 \cdot 2) \tag{1}$$

The procedure-specific risk of HCV infection due to contaminated equipment was calculated for each procedure $p$ as the ratio between the OR of this procedure (denoted $OR_p$) and the injection OR, multiplied by the risk due to injection with contaminated equipment, as follows:

$$Risk_p = \frac{OR_p}{OR_{injection}} \times Risk_{injection} \tag{2}$$

### Individual risk assessment

For each patient, the cumulative risk of HCV acquisition over the entire hospitalisation was computed from the within-hospital individual trajectory, the ward-specific HCV prevalence and procedure-associated risks, as follows:

$$R = 1 - \prod_{i=1}^{n} \prod_{j=1}^{m_i} (1 - r_{j,p} \times P_i \times (1 - A)) \tag{3}$$

where $n$ is the total number of wards visited by the patient, $m_i$ the total number of procedures undergone by the patient in ward $i$, $r_{j,p}$ the risk of HCV acquisition while undergoing the $j^{th}$ procedure if the equipment is contaminated, A the probability of proper equipment handling (i.e., equipment decontamination or use of disposable equipment) and $P_i$ the HCV prevalence in ward $i$.

The risk $r_{j,p}$ was computed as described in the previous section, based on the procedure type $p$. The ward-specific prevalence $P_i$, was used as a proxy of the ward-specific probability of medical equipment being contaminated by HCV prior to infection control procedures. It was considered to be constant over time and equal to the proportion of HCV-positive patients among all patients that passed through ward $i$ in our database. For simplicity, the probability of correct infection control in equipment handling was assumed independent of the procedure type. Syringe reuse was taken as a proxy to estimate this probability at 97%, based on a study by Anwar et al. [26] which found that 3% of nurses from two hospital departments in Egypt and Saudi Arabia reused syringes between patients.

Finally, to maximize statistical power in the identification of hotspots and at-risk profiles, we performed this individual risk assessment using the data from all 500 patients included in the IMMHoTHep study, irrespective of their HCV status upon admission, even though in reality the initially HCV-positive patients were not at-risk.

### Ward-level risk assessment for hotspot identification

To determine the risk of HCV infection associated with each ward in the internal medicine and surgery departments, we calculated for each patient the risk of getting infected through invasive procedures undergone within each unique ward (based on subsets of their trajectories), as in the previous subsection. The distribution of a ward-specific risk was composed of the average risks of all patients visiting it. To shed light on the components of this risk, the ward-level HCV prevalence and average number of procedures per patient and procedure group were also calculated.

### Statistical analyses of patient-level determinants of the HCV infection risk

We investigated differences between the clusters identified through the patient trajectory analyses across: (i) age, HCV infection risk, duration of hospitalization and average number of procedures per patient as quantitative variables; and (ii) gender, education level, marital status, source of admission, source of admission, patient localization, history of hospitalization, hospitalization reason and status at the end of follow-up as categorical variables. Differences were computed using the $\chi^2$ test for qualitative variables and the Kruskal-Wallis test for quantitative variables.

We performed a beta regression to identify upon-admission factors associated with nosocomial HCV risk [27]. As some patients had an infection risk equal to 0, data was transformed following this formula: $y\prime = (y \times (n-1) + 0 \cdot 5)/n$, where $y$ is the risk data and $n$ is the sample size [28]. Explanatory variables included: *age, gender, source of admission, patient localization, history of hospitalization, and hospitalization reason, previous anti-schistosomiasis treatment and history of multiple invasive procedures*. A backward selection was performed to discriminate the best model based on the Akaike information criterion (AIC).

Finally, using logistic regression, we assessed the capacity of a score based on the variables appearing in the best beta regression model to identify at-high-risk patients. We defined high-risk patients as those belonging to the upper 25% of the risk distribution (over the 75[th] percentile). The training data was composed of 70% of the entire dataset whereas the other 30% was used for the testing dataset. If data unbalance was detected, up-sampling was used to equalise sample sizes for both groups. Cross-validation was performed over 50 folds. Area under the ROC curve (AUC), specificity and sensitivity were computed using the R packages *caret* [29] and *Mleval* [30]. A sensitivity analysis was performed on the cut-off for dichotomization; for each case, the Informedness [31] metric was calculated and considered as a proxy of the quality of the model.

### Assessment of patient and ward-focused strategies

We assessed the potential effectiveness of two strategies on the reduction of the HCV infection risk:

i. A patient-focused strategy, assuming the probability *A* of proper equipment handling to be 1 for the most at-risk patients, selected following two sub-strategies: a) randomly (Random-selection) and b) using upon admission the calculated score based on our beta regression model. Here all potential HCV-positive patients within the most-at risk group were considered HCV-negative upon admission, so that they did not impact the risk of other patients visiting the same wards. Strategies targeted at 200 (40%), 150 (30%), 100 (20%), and 50 (10%) at-risk patients among 500 were explored.

ii. A ward-focused intervention, assuming the probability A of proper equipment handling to be 1 within the most at-risk wards. Wards were ranked from higher to lower risk and the

number of targeted wards was chosen based on the cumulative number of patients visiting at least once these particular wards, so that the total number of patients was the closest possible to the number of patients targeted in the corresponding patient-focused scenario.

## Results

### Patient trajectory description

The sequence analysis identified four groups of patients (Fig 1, S2 Fig). Their sizes were heterogeneous (356, 54, 14 and 76 patients, respectively). Group one (the largest one) included patients with a short hospital stay in internal medicine or surgery, Groups two and four represented patients with intermediate lengths of stay in surgery and internal medicine, respectively, and Group three was composed of patients with long stays in internal medicine. In the latter, 36% of patients were deceased at the end of follow-up. Patients in Group three were older than in the other groups (median: 64 years old, IQR [46–67]), had longer hospital stays (20·4 days [17·5–23·2]) and underwent more invasive procedures (median: 43 [15 – 77]) (S1 Table).

### HCV infection risk assessment

The estimated per-procedure median risk of HCV infection due to contaminated equipment ranged from 1·961%% [IQR 1·339% - 2·923%] for endoscopy up to 3·750% [IQR 2·566% - 5·584%] for wound care (S1 Fig).

The median patient HCV infection risk over the entire database was 0·043% [0·026%-0·093%] (Mean: 0·114%, IC95% [0·091%-0·137%]). This risk differed significantly between

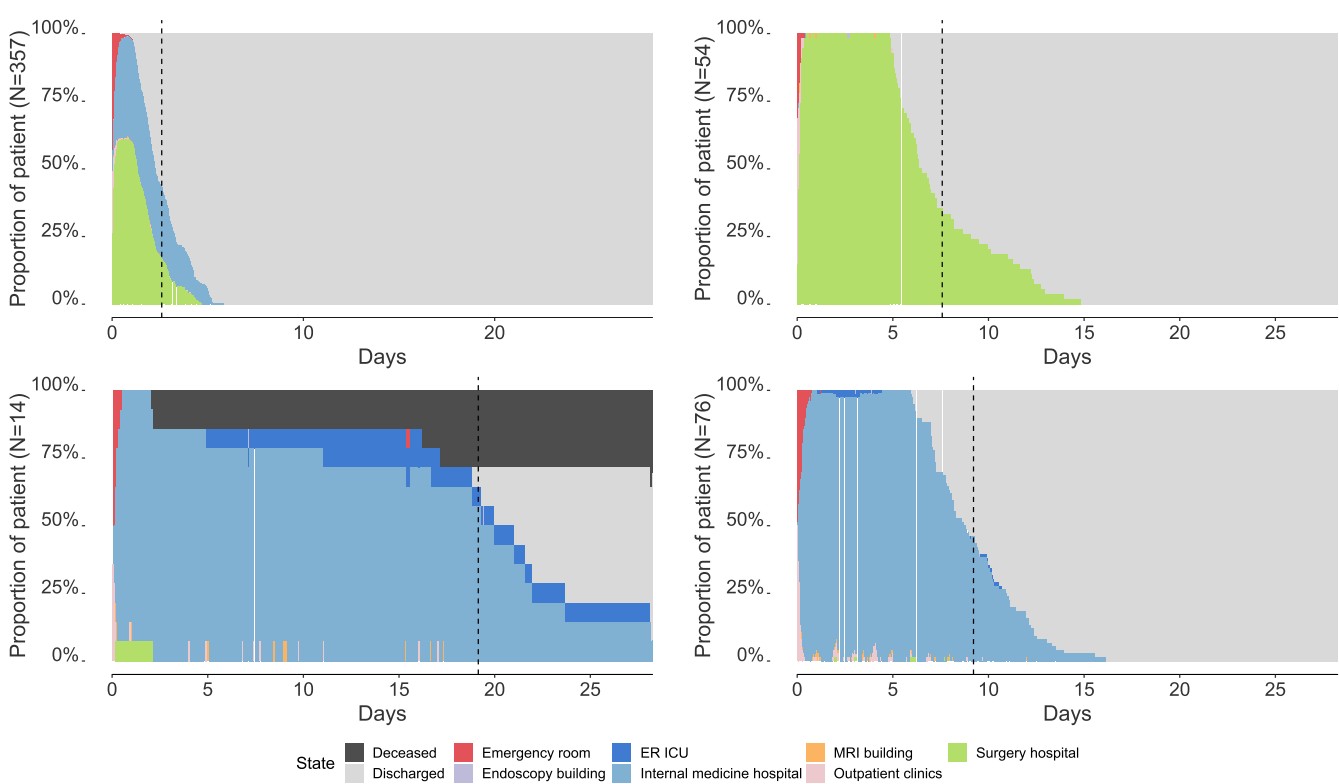

**Fig 1. Chronograms for each of the four clusters of patients identified after sequence analysis.** Dotted lines represent the average length of stay for each group of patients.

patient groups (P<0·001), with a greater risk of getting HCV infected in group 3 (median: 0·470%, IQR [0·081% - 0·823%]) (S1 Table).

## Ward-level risk assessment and hotspot identification

Overall, the risk of HCV infection was higher in the internal medicine hospital compared to the surgery hospital (0·043%, CI 95%: [0·036%-0·050%] vs. 0·188% [0·142%-0·235%], t.test P<$3·62*10^{-9}$).

Within internal medicine, HCV prevalence was found highest in the tropical medicine ward (50% CI 95% [32·100% - 67·900%]), followed by the GIT (31% [15·620%-45·980%]), and geriatric wards (20% [0%-55·060%]) (Fig 2A). Conversely, the average number of procedures within the ER ICU ward was found to be the highest with 33·1 acts, followed by the geriatric ward with 21·6 acts, while the tropical medicine and GIT wards only held the 9th and 10th places among the 15 wards, with 10·45 and 8·24 procedures per patient on average (Fig 2B). The median estimated risk of HCV infection was highest in the geriatric ward (0·621% IQR [0·114%-0·649%], mean: 0·431%) represented by 5 patients, followed by the ward of tropical medicine (0·271% [0·146%- 0·599%], mean: 2·850%), represented by 30 patients and ER ICU ward (0·242% IQR [0·180%-0·811%], mean: 0·474%), represented by 11 patients (Fig 2C).

Within surgery, HCV prevalence was high within the urosurgery, orthopaedics, and neuro-surgery wards, at 20% [0%-55·060%], 57·140% [20·480% -93·800%] and 5·330% [0% -

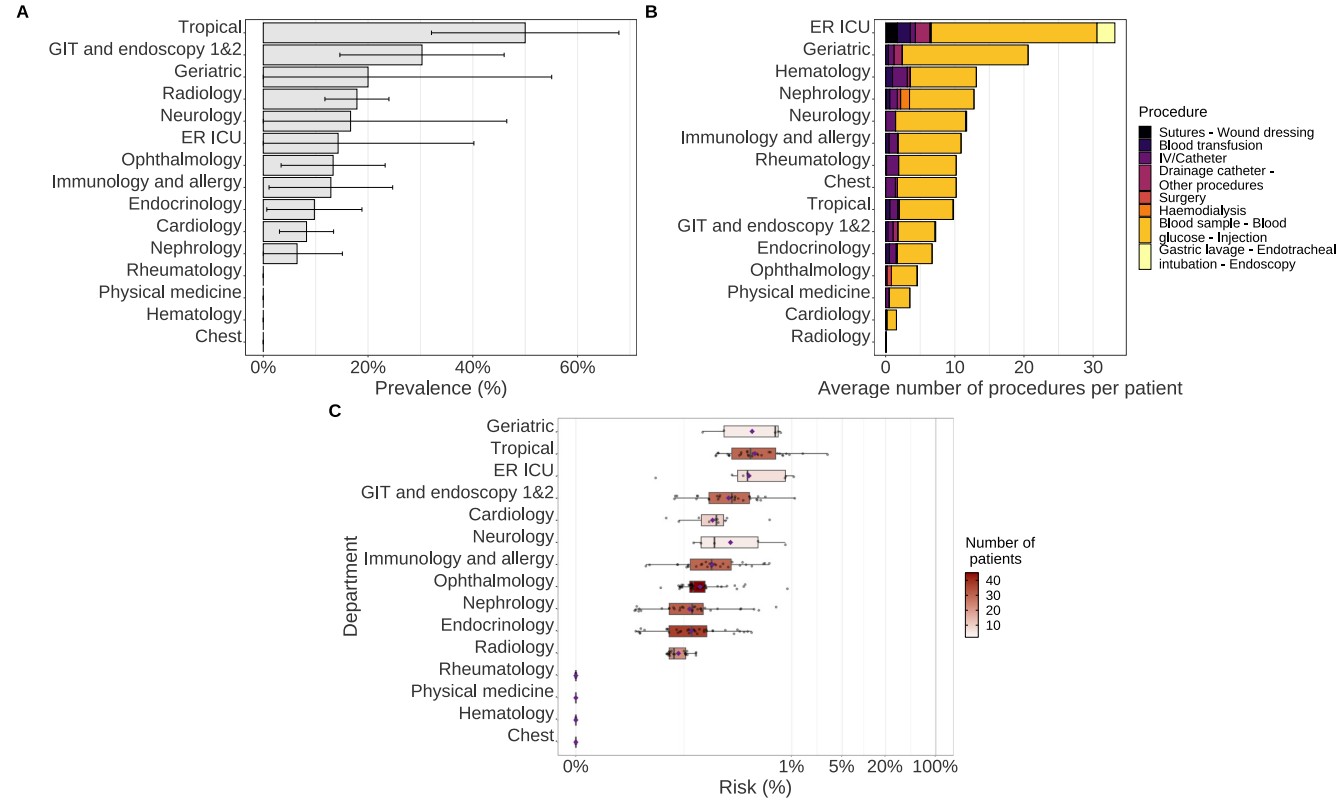

**Fig 2. Panel of ward characteristics for each ward in the internal medicine hospital.** (A) HCV prevalence in each ward with their associated 95% confidence intervals. (B) Average number of procedures per patient. Procedure types are represented from the high-risk ones to the low-risk ones (from left to right). (C) Boxplots of average ward-specific risk of HCV infection, coloured according to the number of patients visiting these wards. Mean values are represented by purple diamond dots.

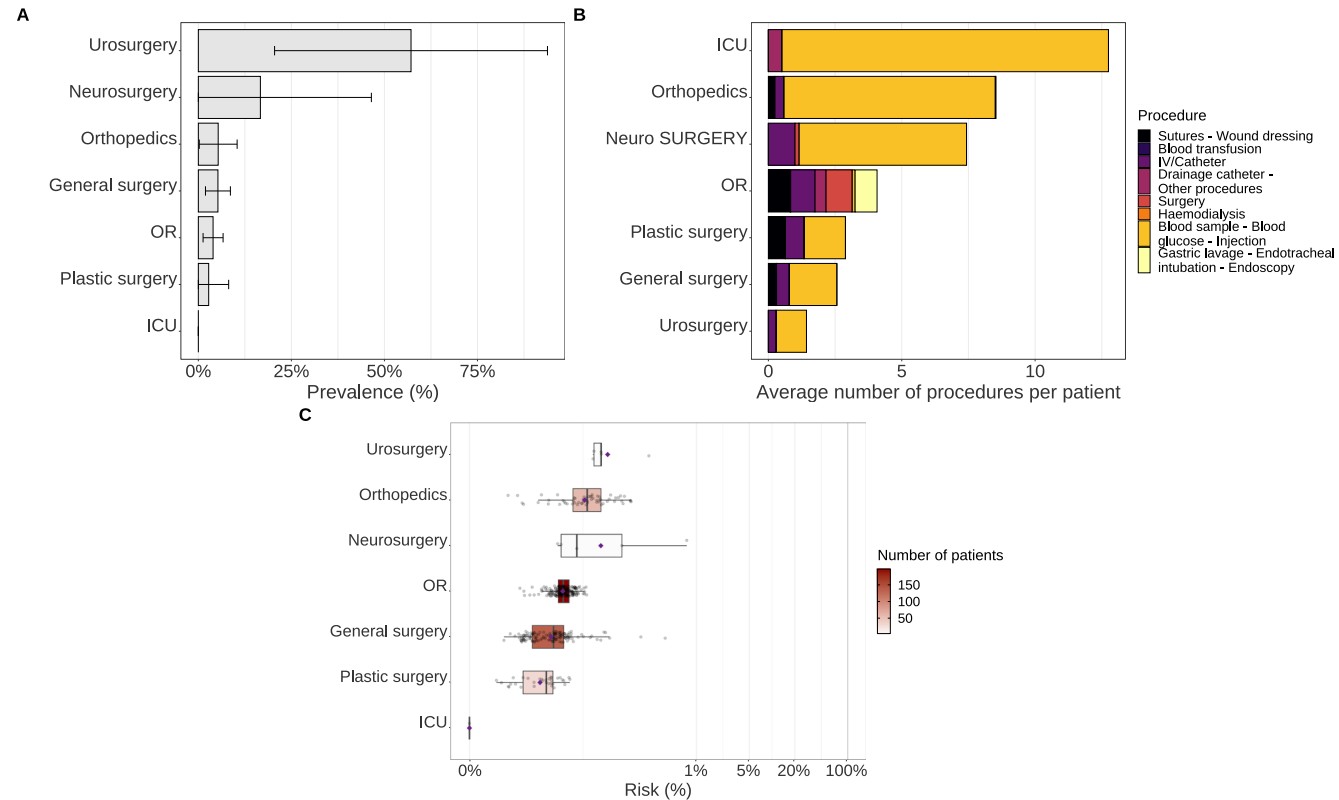

**Fig 3. Panel of ward characteristics for each ward in the surgery hospital.** (A) HCV prevalence in each ward with their associated 95% confidence intervals. (B) Average number of procedures per patient. Procedure types are represented from the high-risk ones to the low-risk ones (from left to right). (C) Boxplots of average ward-specific risk of HCV infection, coloured according to the number of patients visiting these wards. Mean values are represented by purple diamond dots. Three wards are not represented because no patients underwent invasive procedures within them.

15·790%] (Fig 3A). The ICU ward was associated with the highest number of invasive acts, with an average of 12·75 procedures per patient. Within the orthopaedics and neurosurgery wards, patients underwent 8·5 and 7·4 invasive procedures on average, respectively, and only 1·4 in the urosurgery ward (Fig 3B). The highest risk was found in the urosurgery ward (0·045% IQR [0·044%- 0·046%], mean: 0·101%), but only 7 patients visited it. This was followed by the orthopaedics (0·037% IQR [0·024%- 0·053%], mean: 0·046%) and the neurosurgery (0·021% IQR [0·021%- 0·089%], mean: 0·165%) wards, represented by 75 and 6 patients respectively (Fig 3C).

## Identification of at-risk patient profiles upon admission

The best beta regression model explaining the patient HCV infection risk from upon-admission variables is described in Table 2. The hospitalisation cause came out as a key driver of HCV risk, reflecting the higher risk in internal medicine patients, as well as a particularly elevated risk in patients with liver or gastro-intestinal (GIT) complaints. In addition, patients with a history of anti-schistosomiasis treatment were found at higher risk of HCV infection. Other variables selected in the best model were the source of admission and age, with a higher risk in patients admitted via the emergency room or older; and a history of injection or endoscopy.

A score based on these explanatory variables allowed to discriminate high-risk patients upon admission. The calculated AUC was 0·79 (95% CI: [0·71–0·87]) with a sensitivity of 0·73

**Table 2. Result of the multivariate beta-regression analysis.**

| Characteristic | β | Std. Error | p-value |
|---|---|---|---|
| **Source of admission** | | | |
| Outpatient clinic | - | - | |
| Emergency room | 0.081 | 0.051 | 0.114 |
| **Age** | 0.003 | 0.002 | 0.065 |
| **Reason for hospitalisation** | | | |
| General surgery | - | - | |
| Special surgery | 0.064 | 0.075 | 0.395 |
| General IM | 0.276 | 0.089 | <0.01** |
| Special IM | 0.195 | 0.068 | <0.01** |
| Liver/ GIT complaint | 0.443 | 0.092 | <0.001*** |
| **Previous anti-schistosomiasis treatment** | | | |
| No / Doesn't remember | - | - | - |
| Yes | 0.169 | 0.084 | 0.043* |
| **Previous injection** | | | |
| No / Doesn't remember | - | - | - |
| Yes | 0.097 | 0.052 | 0.062 |
| **Previous endoscopy** | | | |
| No / Doesn't remember | - | - | - |
| Yes | 0.118 | 0.072 | 0.102 |

IM: internal medicine. GIT: gastro-intestinal

[0·65–0·79], and a specificity of 0·68 [0·54–0·79] (S3 Table). Based on a sensitivity analysis, we found that using a cut-off at the 90[th] percentile of the overall distribution led to the best logistic regression based on the Informedness criteria (S2 Table).

## Assessment of patient and ward-focused strategies

All simulated interventions focusing on at least 20% of patients led to at least a two-fold reduction of the overall risk, except those based on randomly selected patients (Fig 4, S3 Table) In addition, patient-focused interventions were generally found to be more effective than ward-focused (Fig 4, S3 Table). Nevertheless, for interventions targeting 20% (100) of patients and less, focusing on the most at-risk wards was more efficient at reducing the risk than score-based patient targeting.

## Discussion

This study aimed at better understanding patient trajectories within an Egyptian hospital to help manage the HCV infection risk. Our work was based on data collected on 500 patients within Ain Shams hospital, Egypt, and on a meta-analysis investigating the risk of HCV infection for multiple hospital-based procedures [24], from which we computed HCV infection risks for all patients over the course of their hospitalisation.

While we estimated a low overall HCV infection risk, we found that some upon-admission patient characteristics were related to a higher risk: age, reason of hospitalization, and history of previous invasive procedures. We proposed a score to detect high risk patients upon their admission and assessed the effect of simulated interventions on the overall risk of HCV infection during hospitalization. Selecting patients using our score was always more effective than randomly selecting patients upon-admission.

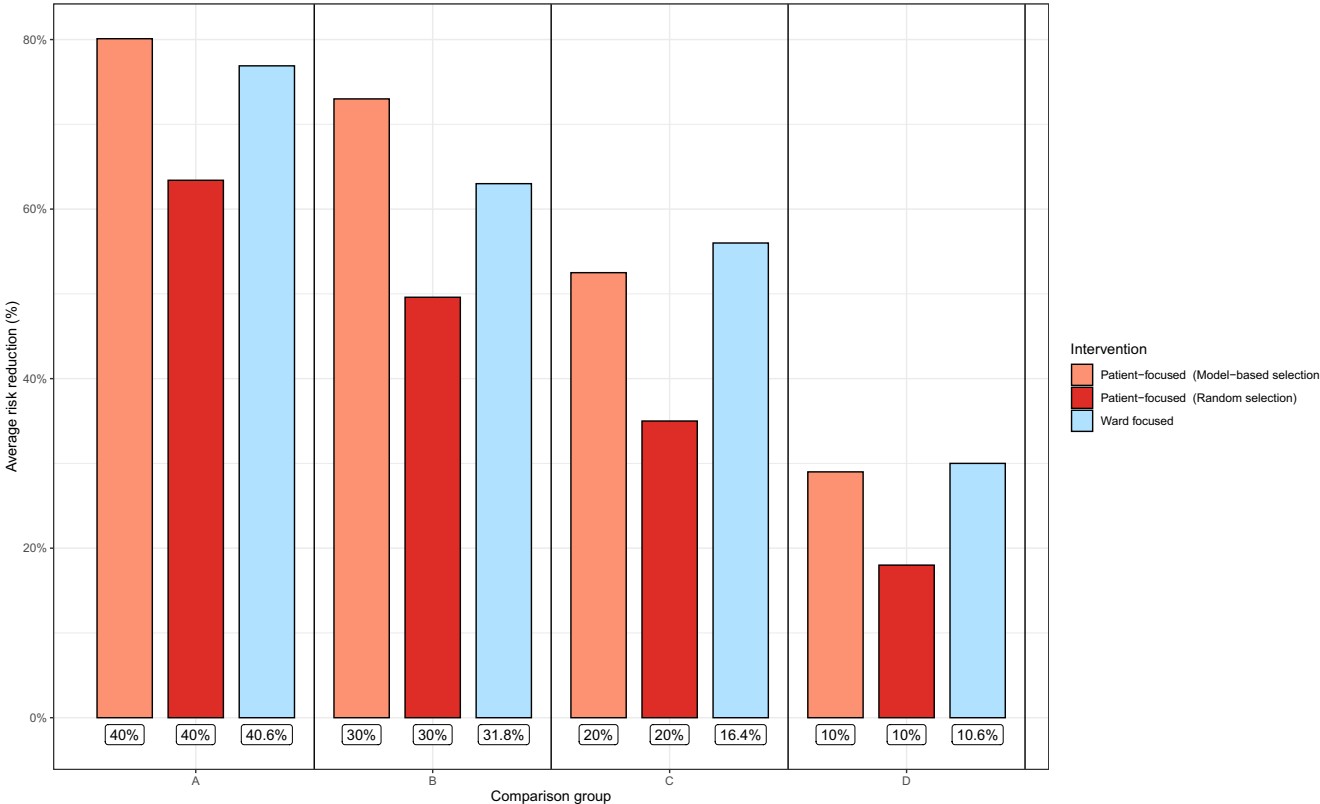

**Fig 4. Average effect of simulated intervention on the overall risk of HCV infection during hospitalization.** Labels under bars correspond to the proportion of concerned patients for a given intervention for the four sub-scenarios considered in the analysis (Comparison groups A, B, C and D). As proportions of patients for the ward-focused scenario were chosen based on the number of cumulative patients in these wards, they were not exactly equal to the proportions given for patient-based scenarios.

Due to high uncertainty, notably on per-procedure infection control practices, our risk estimates should be considered relatively, rather than focusing on their absolute values. For instance, we propose a prioritisation of wards in terms of HCV infection risk in the surgery and internal medicine departments. In particular, we found that the internal medicine department is the most at-risk of HCV infection, with the geriatric, tropical and endoscopy wards within this department identified as potential "hotspots". This is consistent with a previous risk assessment study conducted in a German hospital in 2008 in which the highest risk of bloodborne pathogen infection for HCW was within the internal medicine departments [32].

Our results might not be generalizable outside Egypt, as the epidemiology of HCV in this country is unique. In particular, in our database, most of the infected patients were older than non-infected ones and most of them had a history of anti-schistosomiasis treatment performed during the mass treatment campaign initiated by the government between 1950 and 1980, which led to a massive diffusion of HCV within the country [33]. While this may impact both the hotspots we identified within the hospital and the at-risk patient profiles we determined, the approach we propose to assess the HCV infection risk may still be extended to other settings and contexts beyond Egypt, provided the necessary data is available.

To the best of our knowledge, only a few earlier studies have attempted to assess the individual risk of HCV infection for patients and healthcare workers within hospitals. Among these studies, only two described models investigating the patient-to-patient transmission risk

using a probabilistic approach [14, 15]. The first one [14] described the individual risk for patients hospitalized in a haemodialysis unit and the other one quantified the risk of blood-borne pathogen infection during an invasive procedure [15]. Although these studies used the HCV prevalence level within wards to estimate the HCV infection risk for hospitalized patients, none used detailed longitudinal data to investigate the individual infection risk during hospitalization. In addition, a few studies directly reported HCV incidence in hospitalized patients. However, all were focused on haemodialysis units [34–36]. Finally, as far as we know, no earlier study has investigated the impact on HCV infection risk of patient-based and ward-based interventions similar to those proposed in our work. Some studies investigated the effect of increased prevention and control in healthcare workers on the incidence of occupational exposure to bloodborne pathogens [37, 38] but none seemed to focus on the effect of these prevention measures on the HCV incidence in patients.

Several data-related and methodological limitations could be highlighted in our study.

First, we were limited by our data and had few observations of patients for multiple wards. In particular, the highest estimated risks were observed in wards that received a low number of patients (Neurosurgery and geriatric). In future studies, investigating these specific risks based on more patient visits might give more accurate estimates of the corresponding ward-level HCV infection risks.

Second, we did not have access to HCV status upon patient discharge. A study assessing the HCV status after hospitalisation in addition to the HCV status upon admission would be needed to really quantify the risk of getting HCV infected during a hospital stay from data. However, we believe that the analyses we performed, based on detailed data on patient trajectories and per-procedure risk ranking, do provide valid conclusions in terms of the identification of possible hotspots and at-risk patient profiles within the hospital.

Third, we only accounted for transmission between patients within the same ward and did not investigate potential transmission from healthcare workers to patients, which can be another HCV gateway [39]. This may have led us to under-estimate all the HCV acquisition risks, but should not have affected the prioritisation we propose in terms of geographical hotspots and patient profiles upon admission. Indeed, an earlier study performed in the same hospital on a larger staff population confirmed that HCV RNA positive healthcare workers were very rare, and that highest proportions of HCV-infected healthcare workers were found in the internal medicine department [40].

Fourth, our beta-regression model identified several upon-admission variables as associated with the risk of HCV infection during hospitalization. These associations should not be interpreted as causal. For example, patients with a history of anti-schistosomiasis treatment were found to be more at risk of HCV infection, possibly reflecting the high prevalence among other patients they are exposed to.

In a context of limited budget and human resources, this work may help better manage the HCV risk within Egyptian hospitals in two ways. First, infection control could be reinforced locally in the hotspots we identified. This could for instance imply systematic HCV screening for patients newly admitted to these specific wards, hiring of dedicated hygiene personnel within these wards, or allocation of the available disposable equipment to these wards. Second, the score we proposed could be systematically computed upon admission for all newly admitted patients. Those identified as at high risk could then be "flagged" for reinforced precautions over the course of their hospitalisation. When comparing such ward-focused and patient-focused strategies, we found that interventions targeted at identified at-risk patients upon admission were most effective. However, interventions focused on ward hotspots also allowed to reduce the risk more than two-fold and may in practice prove both easier to implement logistically and more acceptable from an ethical point of view. In addition, when hospital

resources only allowed to target less than 20% of patients for reinforced infection control, ward-focused interventions were actually most effective.

Even if the Egyptian government implemented a large HCV test and treat campaign in 2018 that led to a significant prevalence reduction, these results may still help to reach more easily WHO HCV elimination objectives by pointing out the most at risk patients and wards. In addition, the framework we developed could be extended to assess and manage iatrogenic HCV risks in other hospitals or risks associated with other blood-borne pathogens such as HIV or HBV. This would require, in the first case, the collection of data on patient trajectories similar to the IMMHoTHep data in other hospitals; and in the second case, estimates of the per-procedure infection risks associated with these other pathogens.

Finally, in future work, the detailed data we collected on patient trajectories and invasive procedures could be used to inform mechanistic models simulating dynamically the transmission of HCV or other blood-borne pathogens within the hospital. Such models would allow to assess the impact of potential control measures in a more accurate way than the very simplified assessment proposed here.

## Supporting information

**S1 Checklist. Inclusivity in global research.**
(DOCX)

**S1 Table. Characteristics of each group found after sequence analysis.**
(DOCX)

**S2 Table. Summary of the impact of patient and ward-focused strategies on the risk of HCV infection during hospitalization.**
(DOCX)

**S3 Table. Sensitivity analysis for the cut-off value of the risk considered in the logistic regression.**
(DOCX)

**S1 Fig. Distributions of the procedure–specific risks of HCV infection in case of contaminated equipment.**
(JPG)

**S2 Fig. Point Biserial Correlation (PBC) for 1 to 20 clusters.** PBC was very similar for 3, 4 and 5 partitions. Therefore, we chose to build 4 clusters of patients (vertical dashed line).
(PNG)

**S1 Text. Mode calculation.**
(DOCX)

## Author Contributions

**Conceptualization:** Paul Henriot, Kévin Jean, Laura Temime.

**Data curation:** Wagida A. Anwar, Maha El Gaafary, Samia Abdo, Mona Rafik, Wafaa M. Hussein, Dalia Sos, Isis Magdy.

**Formal analysis:** Paul Henriot.

**Funding acquisition:** Paul Henriot.

**Methodology:** Paul Henriot, Kévin Jean, Laura Temime.

**Supervision:** Kévin Jean, Laura Temime.

**Validation:** Wagida A. Anwar, Kévin Jean, Laura Temime.

**Visualization:** Paul Henriot.

**Writing – original draft:** Paul Henriot.

**Writing – review & editing:** Paul Henriot, Wagida A. Anwar, Maha El Gaafary, Samia Abdo, Mona Rafik, Wafaa M. Hussein, Dalia Sos, Isis Magdy, Kévin Jean, Laura Temime.

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
