## [Decision Letter · Decision Letter 0]

5 Dec 2023

PGPH-D-23-01725

Preventing iatrogenic HCV infection: A quantitative risk assessment based on observational data in an Egyptian hospital.

Dear Dr. Henriot,

Thank you for submitting your manuscript to PLOS Global Public Health. After careful consideration, we feel that it has merit but does not fully meet PLOS Global Public Health’s publication criteria as it currently stands. Therefore, we invite you to submit a revised version of the manuscript that addresses the points raised during the review process.

Provide more clarity in the methods on how your findings are linked to prevention of Hepatitis C.

Improve your discussion to align it to your findings, citing more literature about prevention of iatragenic hepatitis C. 

Revise your abstract to make it more structured. 

Respond to all other reviewer comments 

We look forward to receiving your revised manuscript.

Kind regards,

Reuben Kiggundu

Academic Editor

Journal Requirements:

If you are reporting a retrospective study of medical records or archived samples, please ensure that you have discussed whether all data were fully anonymized before you accessed them and/or whether the IRB or ethics committee waived the requirement for informed consent. If patients provided informed written consent to have data from their medical records used in research, please include this information."

3. Please provide separate figure files in .tif or .eps format.

Additional Editor Comments (if provided):

Reviewers' comments:

Reviewer's Responses to Questions

**Comments to the Author**

1. Does this manuscript meet PLOS Global Public Health’s publication criteria? Is the manuscript technically sound, and do the data support the conclusions? The manuscript must describe methodologically and ethically rigorous research with conclusions that are appropriately drawn based on the data presented.

Reviewer #1: Yes

Reviewer #2: Yes

2. Has the statistical analysis been performed appropriately and rigorously?

Reviewer #1: Yes

Reviewer #2: Yes

3. Have the authors made all data underlying the findings in their manuscript fully available (please refer to the Data Availability Statement at the start of the manuscript PDF file)?

Reviewer #1: Yes

Reviewer #2: Yes

4. Is the manuscript presented in an intelligible fashion and written in standard English?

Reviewer #1: Yes

Reviewer #2: Yes

5. Review Comments to the Author

Reviewer #1: 1. Congratulations to the research team for their interesting study which addresses risk and how to prevent iatrogenic HCV infection in hospitals in Egypt.

2. The abstract should be more concise and informative, highlighting the main objectives, methods, results and implications of the study.

3. The introduction lacks detailed background information on HCV infection in Egypt (epi-data, morbidity and mortality, gaps in knowledge and why the objectives/aims of this study). If there is paucity of data on HCV infection in healthcare in Egypt, please indicate so. Lines 305-318 could be moved to the introduction.

4. Methodology should include more information on data collection procedures. Lines 70-72 should be rephrased to ensure that the 15 and 10 wards are respectively referring to medical and surgical departments.

5. The findings and limitations were well presented. How does these limitations affect the accuracy and generalizability of the findings?

6. The discussions of the findings is limited and should be expanded to include the major findings of the study as well as similar studies on HCV nosocomial infections to indicate similarities and differences in the findings and how they should be interpreted.

Reviewer #2: "Preventing iatrogenic HCV infection: a quantitative risk assessment based on observational data in Egyptian Hospitals " is a very interesting research study to share. The authors were detailed in methodology and results. However, the discussion could not figure out at risk patients' profile before admission and the assessment of patients and ward focus strategies. the at risk patients in other studies were not compared with this study. Similarly, the assessment of patients and ward focus interventional strategies were not substantiated: were they the first of its kind reported or any comparison with other similar intervention. The limitations were fully stated and clear. This is an impressive manuscript if these observed parts were considered for discussion.

6. PLOS authors have the option to publish the peer review history of their article (what does this mean?). If published, this will include your full peer review and any attached files.

**Do you want your identity to be public for this peer review?** For information about this choice, including consent withdrawal, please see our Privacy Policy.

Reviewer #1: No

Reviewer #2: **Yes: **Sikiru Olanrewaju Badaru

---

## [Editor Report · Decision Letter 1]

30 Jan 2024

Preventing iatrogenic HCV infection: A quantitative risk assessment based on observational data in an Egyptian hospital.

PGPH-D-23-01725R1

Dear Dr. Henriot

We are pleased to inform you that your manuscript 'Preventing iatrogenic HCV infection: A quantitative risk assessment based on observational data in an Egyptian hospital.' has been provisionally accepted for publication in PLOS Global Public Health.

Best regards,

Dr. Reuben Kiggundu

Academic Editor
